# MnO_2_@Corncob Carbon Composite Electrode and All-Solid-State Supercapacitor with Improved Electrochemical Performance

**DOI:** 10.3390/ma12152379

**Published:** 2019-07-26

**Authors:** Xin-Sheng Li, Man-Man Xu, Yang Yang, Quan-Bo Huang, Xiao-Ying Wang, Jun-Li Ren, Xiao-Hui Wang

**Affiliations:** State Key Laboratory of Pulp and Paper Engineering, South China University of Technology, Guangzhou 510640, China

**Keywords:** natural biomass, porous carbon, architecture, MnO_2_, supercapacitor

## Abstract

Two corncob-derived carbon electrode materials mainly composed of micropores (activated carbon, AC) and mesopores/macropores (corncob carbon, CC) were prepared and studied after the anodic electrodeposition of MnO_2_. The capacity of the MnO_2_/activated carbon composite (MnO_2_@AC) electrode did not noticeably increase after MnO_2_ electrodeposition, while that of the MnO_2_/corncob carbon composite (MnO_2_@CC) electrode increased up to 9 times reaching 4475 mF cm^−2^. An asymmetric all-solid-state supercapacitor (ASC) was fabricated using AC as the anode, MnO_2_@CC as the cathode, and polyvinyl alcohol (PVA)/LiCl gel as the electrolyte. An ultrahigh specific capacitance of 3455.6 mF cm^−2^ at 1 mA cm^−2^, a maximum energy density of 1.56 mW h cm^−2^, and a long lifetime of 10,000 cycles can be achieved. This work provides insights in understanding the function of MnO_2_ in biomass-derived electrode materials, and a green path to prepare an ASC from waste biomass with excellent electrochemical performance.

## 1. Introduction

Carbonaceous materials derived from abundant waste biomass (sugarcane bagasse, potato, lotus seedpod, rice husk, taro epidermis, banana peel, mulberry leaves, coffee grounds, etc.) [1,2,3,4,5,6,7,8] are considered as good electrode materials for supercapacitors due to their high specific surface area, good conductivity, easy accessibility, low cost, and renewability. However, biomass-based supercapacitors suffer from slow charge/discharge performances and a limited energy density because of their micropore structure, which limit their application in high-performance supercapacitors [9]. 

It has been recognized that a rational microstructure design and the inclusion of active materials are two effective approaches to improve the capacitance performance of biomass-derived carbon materials [10,11,12]. Microstructure design mainly pays attention to constructing a hierarchical pore structure of the carbon material using improved pretreatment and carbonization processes [13], while the other strategies concentrate on the introduction of pseudocapacitive materials such as heteroatoms [14,15], conducting polymers [16,17], transition metal sulfides [18], and oxides [19,20,21,22,23,24]. Among these pseudocapacitive materials, manganese dioxide (MnO_2_) is the most promising one due to its high theoretical specific capacitance (1370 F g^−1^), wide potential window, rich reserves, and being environmentally friendly [25]. 

Anchoring MnO_2_ on carbon substrates such as activated carbon (AC) using electrochemical deposition offers an effective way to obtain a superior capacitive property and rate capability [26,27,28,29,30,31]. Zhang et al. prepared an electrodeposited MnO_2_@AC composite air cathode with a power density of 1554 mW m^−2^, which was 1.5 times that of an AC air cathode [32]. In some other reports, the mass-specific capacitance of MnO_2_@AC electrode was raised by 105.8% [33] or even three times [34] that of AC electrode. However, the application of the MnO_2_ functionalized biomass-derived carbon materials in all-solid-state supercapacitors (ASC), which are essential for the next-generation portable electronics, still remains a challenge due to limited performance of the device. 

In this work, high performance electrode materials were prepared from corncob carbon using the carbonization and electrodeposition of MnO_2_. It was interesting to find that the performance of corncob carbon (CC) without chemical activation was significantly improved after MnO_2_ deposition. The obtained MnO_2_@CC electrode showed a high area-specific capacitance of 4475 mF cm^−2^, which was 9 times that of CC. Highly porous and conductive corncob AC was also obtained, although its performance could not be greatly improved via MnO_2_ deposition. A sandwich-structured ASC utilizing corncob AC as an anode, MnO_2_@CC as a cathode, polyvinyl alcohol (PVA)/LiCl gel as an electrolyte, and a thin sulfonated membrane as separator was fabricated. The device demonstrates high power/energy density and shows great potential in green and renewable energy storage for its reliability and good biocompatibility. 

## 2. Materials and Methods 

### 2.1. Materials

The corncob raw materials were collected in Henan province in the middle of China. Polytetrafluoroethylene (PTFE) solution was purchased from Aladdin Chemistry Co., Ltd. (Shanghai, China). A membrane filter (MPF50AC) separator was purchased from Nippon Kodoshi Plant of Japan (Kochi-ken, Japan). Acetylene black and nickel foam were obtained from Lili Lithium Electric Technology Center (Taiyuan, China). Hydrochloric acid (HCl) and potassium hydroxide (KOH) were analytical grade chemical reagents used without purifying. Manganese acetate (Mn(CH_3_COO)_2_) and lithium chloride (LiCl) were provided by the Fuchen Chemical Reagent Factory (Tianjin, China). Sodium sulfate (Na_2_SO_4_) was purchased from Runjie Chemical Reagent Company (Shanghai, China). We purchased polyvinyl alcohol (PVA-124) from Sinopharm Chemical Reagent Co., Ltd (Shanghai, China); it had a molecular weight of 105,000, degree of polymerization in the range from 2400 to 2500, and a hydrolysis degree of 98–99%.

### 2.2. Synthesis of AC and CC 

AC was prepared using one-step carbonization. The dried corncob was chopped into 5–10 cm pieces using a disintegrator. After that, solid KOH was mixed with corncob residues at a KOH/carbon mass ratio of 0.5 and a small amount of water was added to just cover the corncob residues. After being soaked in KOH solution for 10 h, the corncob residues were dried at 80 °C for 12 h and further carbonized at 750 °C for 1.5 h with a heating rate of 3 °C min^−1^ in an argon atmosphere. The carbonized samples were washed several times using 1 M HCl and deionized (DI) water, then vacuum-dried overnight. Finally, the corncob-based porous carbons could be created.

CC was prepared without the activation of KOH. The dried natural corncob was cut into the desired size (2 cm × 1 cm) along the axial direction and then pre-carbonized at 300 °C for 1 h with a heating rate of 5 °C min^−1^ in an argon atmosphere, followed by carbonization at 900 °C for 1 h with a heating rate of 6.7 °C min^−1^. Then, the obtained solid biochar was washed successively using DI water and the moisture was removed to obtain the final CC.

### 2.3. Preparation of AC and CC Electrodes

The AC electrode was prepared as follows. The AC pieces were ground into AC power in an agate mortar. Then, 1 g AC power was added to 30 mL water and the mixture solution was stirred magnetically for 0.5 h at ambient temperature. A homogeneous dispersion was obtained by sonicating the mixture for several hours. A total of 0.125 g carbon black and 0.125 g PTFE (mass ratio of AC/carbon/PTFE = 8:1:1) were added into the beaker under magnetic stirring and sonicating in order. After that, the mixture was dried into a viscous slurry at 100 °C in the vacuum oven. Finally, the mixed slurry was pressed into a membrane and pasted on the nickel foam (1 cm × 2 cm), followed by vacuum drying to get the final AC electrodes.

The cleaned CC were used as the electrode directly due to its self-supporting and excellent electrical conductivity.

### 2.4. Preparation of MnO_2_@AC and MnO_2_@CC Electrodes

MnO_2_ was grown in situ on an AC electrode with a typical anodizing method performed in the electrochemical station (CH Instruments Ins, Shanghai, China). Electrodeposition was achieved by using a platinum (Pt) plate and Ag/AgCl electrode as a counter electrode and reference electrode [34], respectively. Before electrodeposition, the obtained AC electrode (1 cm × 1 cm) was soaked in the electrolyte containing 0.1 M of Mn(CH_3_COO)_2_ and 0.1 M of Na_2_SO_4_ for one night. A constant current of 1 mA/cm^2^ was applied to ensure the growth of MnO_2_ nanosheets on the AC membrane and its channels when the deposition time changed from 5 min to 2 h. Next, the as-prepared samples were rinsed gently by the water and vacuum dried at 60 °C to obtain the final MnO_2_@AC electrodes.

The preparation of MnO_2_@CC electrodes was similar to that of MnO_2_@AC electrodes except for the difference that the CC block was clamped by a couple of platinum sheets, which made sure there was a uniform distribution of current to support the growth of the MnO_2_ nanosheets. 

### 2.5. Fabrication of All-Solid-State ASCs

The PVA/LiCl gel electrolyte was prepared as follows: 6.3 g of LiCl and 30 mL of DI water were added in the beaker and stirred for 5 min to get the transparent solution. Next, 3 g PVA was added into the solution, followed by an oil bath with magnetic stirring at 90 °C for 1 h to dissolve the PVA.

In a typical assembling process of ASCs, the AC anode, MnO_2_@CC cathode, and membrane filter were soaked in the PVA/LiCl electrolyte for 5 min and then coagulated in a vacuum oven at room temperature overnight. After that, the ASCs were constructed by pressing the AC anode and MnO_2_@CC cathode together with a membrane filter (MPF50AC) between them.

### 2.6. Material Characterization

The microstructure of these samples were investigated using scanning electron microscopy (SEM, ZEISS Merlin, Oberkochen, Germany) and transmission electron microscopy (TEM, JEM-2100, JEOL Ltd., Tokyo, Japan). The crystalline structure of the electrode materials was determined using an X-ray diffractometer (XRD, D/max-IIIA with nickel-filtered Cu Kα radiation (λ = 0.15418 nm), Rigaku Corporation, Tokyo, Japan). X-ray photoelectron spectroscopy (XPS) was conducted on an AMICUS (Shimadzu, Japan) spectrometer with monochromated Mg Kα radiation. The nitrogen sorption measurements were carried out using an ASAP 2020 analyzer (Micromeritics, Georgia, GA, USA). The Raman spectra were recorded on a Raman Spectrometer (HORIBA Jobin Yvon, Paris, France).

### 2.7. Electrochemical Characterization

The electrochemical behavior of a single electrode was studied using a three-electrode mode in 1 M Na_2_SO_4_ aqueous solution. Cyclic voltammetry (CV) measurements were carried out to evaluate the capacitive behavior of the AC and CC. Galvanostatic charge/discharge (GCD) curves were tested to calculate the specific capacitance of AC and CC. An external window voltage of −1 V to 0 V and 0–0.8 V, which correspond to AC- and CC-based electrodes, respectively, were applied in the charging and discharging process. Cycling stability was tested using a land-battery texting system.

## 3. Results

### 3.1. Morphological Analysis

Herein, two different porous carbon materials: AC and CC were prepared from the agricultural byproduct corncob, as shown in Scheme 1. AC was prepared using KOH activation and carbonization of chopped corncob pieces, while CC was obtained via direct carbonization of chopped corncob pieces without activation. The morphology of natural corncob tissue, AC, and CC are presented in Appendix A. The natural corncob has a rich vascular tissue with tracheids and vessels for transporting water and nutrients, which endows it with natural open channels. Appendix A shows the SEM images of the natural corncob, where 3D vertical channels (10–30 μm) can be observed. With KOH activation, the original geometry of the natural corncob was destructed. A loose and highly porous structure with a large amount of micropores was created (Appendix A). Unlike AC, the original vertical channel structure of the natural corncob could be well-kept in CC, whose channel diameter was between 30 to 60 μm (Appendix A). Besides the major tunnels, there were also many small channels (1–3 μm) on the inner wall of the macroporous channels (Appendix A), which constituted a hierarchical pore structure. In most cases, the micropores endow the material with a high specific surface area and large specific capacitance, while the mesopores and macropores can facilitate the transportation of electrolytes [35]. 

To improve the capacitive performance of AC and CC, a constant current of 1 mA cm^−2^ was applied to them to grow MnO_2_ in situ. Figure 1 shows the morphology and structure of AC and CC with electrodeposited MnO_2_ for different durations. Due to the high conductivity of AC and CC (0.02 Ω/sq), MnO_2_ was easily electrodeposited in the substrate. Figure 1a shows that the pores in AC had been filled with MnO_2_ crystals with only 5 min of deposition, which induced a blocking of channels and the destruction of pore structures. The deposited MnO_2_ crystals presented a nanoflower shape (Figure 1b), which is similar to previous reports [36,37,38]. With a prolonged deposition time, the MnO_2_ crystals grew into a more uniform and smooth film (Appendix A). After 120 min, a very flat and continuous MnO_2_ crystals film had been obtained, while the crystal shape was almost unchanged (Appendix A). 

On the other hand, as Figure 1c shows, the pore structure of CC was well-preserved in MnO_2_@CC. Appendix A shows the morphology of MnO_2_@CC with increased deposition time from 30 min to 10 h. As the deposition time increased, the thickness of the channel wall increased gradually, resulting in a reduction of the channel diameter. After manganese dioxide was electrodeposited on CC for 2 h, the pore wall became thicker (10 μm), but the vertical channels were still unimpeded. With a layer of manganese dioxide nanosheets (Figure 1d) growing on the inner and outer surfaces of the pore wall, the original small pore on the inner pore wall disappeared. When the electrodeposition lasted up to 5 h, the MnO_2_ crystals penetrated throughout the tunnels, while the surface of MnO_2_@CC still remained unimpeded according to the cross-sectional view image (Appendix A). When the deposition time reached up to 10 h (Appendix A), stacked MnO_2_ crystals separated by criss-crossed cracks appeared on the surface, probably due to the inhomogeneous nuclei growth. In summary, a very short electrodeposition time (5 min) led to the clogging of pores in MnO_2_@AC, while MnO_2_@CC could keep the original macroporous structure over a long electrodeposition time of up to 2 h. The difference in the growth rate of MnO_2_ crystals in the two materials was probably determined by their different surface areas. Smaller pore structures and larger specific surface area (SSA) lead to a higher current density and the faster growth of MnO_2_ crystals.

The detailed microstructure of the electrodeposited MnO_2_ in MnO_2_@CC-2 h was characterized using high resolution TEM. Figure 1e shows that the individual MnO_2_ nanoflower was composed of interconnected sheet-like subunits with a thickness of several nanometers. Meanwhile, it can be seen that there were some small disordered or defective regions and spaces ranging from 5 nm to 14 nm on the MnO_2_ nanosheets (red circles in Figure 1f). The corresponding selected-area electron diffraction (SAED) pattern (inset of Figure 1f) demonstrated that the deposited MnO_2_ was polycrystalline. Furthermore, the lattice spacings of 0.7 nm and 0.24 nm correspond to the interplanar distances of (001) and ($\bar{1}11$) planes of δ-MnO_2_, respectively. XRD results confirmed the this (Appendix A). As shown, the two broad peaks around 23° and 43.6° correspond to the (002) and (100) diffraction planes of porous carbon, respectively. Apart from them, new peaks appeared at 36.7° and 65.7°, which correspond to the diffraction of ($\bar{1}$11) and (311) planes of δ-MnO_2_. The weak intensity and the broad width of two peaks indicate the poor crystallinity of layered δ-MnO_2_.

To further demonstrate the change in microstructure of the corncob-based materials after depositing MnO_2_, the nitrogen adsorption–desorption isotherms and pore size distribution using the Barrett–Joyner–Halenda (BJH) method were investigated. According to Figure 2a–d, the isotherms of these porous materials all present typical type IV isotherms and the pore size distribution curves confirm the hierarchical porosity. The sharp increase (Figure 2b) at a low relative pressure (P/P^0^ = 0.06–0.4) is characteristic of micropores. AC contained far more micropores than MnO_2_@AC. The specific surface area (S_BET_) of AC decreased from 626.08 m^2^ g^−1^ to 32.76 m^2^ g^−1^ after the MnO_2_ electrodeposition. These results prove that the micropores were blocked after MnO_2_ was electrodeposited. The appearances of hysteresis demonstrated the existence of mesopores. Meanwhile, a sharp upward tendency (Figure 2d) at high partial pressures (P/P^0^ = 0.4–1.0) signified that there were many macropores in CC and MnO_2_@CC. The S_BET_ of MnO_2_@CC was 3.5 m^2^ g^−1^, which was close to that of CC (5.2 m^2^ g^−1^). This result indicates that the macropores structure in CC was not obviously affected by MnO_2_ loading. Figure 2e,f graphically illustrates the architecture of AC and CC after electrodepositing MnO_2_. For AC, MnO_2_ crystals were grown not only on the surface but also inside the micropores, even with a very short electrodeposition time of 5 min, resulting in a very clogged microstructure. For CC, due to its macroscopic pore structure, the deposition of MnO_2_ was much slower and did not significantly change its microstructure.

### 3.2. Element Analysis and Surface Chemistry Characterization

Appendix A represents the Raman spectrum of the corncob-derived materials. The two peaks located at 1350 cm^−1^ and 1580 cm^−1^ belong to the D and G bands, respectively. The G band was the in-plane vibration of graphitic carbon while the D band was ascribed to structural defects [39]. The I_D_/I_G_ intensity ratio provides the degree of disorder and the average size of the sp^2^ domains. The I_D_/I_G_ ratio of AC, MnO_2_@AC, CC, MnO_2_@AC were 0.87, 0.93, 1.00, and 0.97 respectively, which means they all had typical amorphous carbon structures. The I_D_/I_G_ ratio of CC was higher than AC, which may be due to the higher carbonization temperature. 

The XPS spectra of MnO_2_@AC and MnO_2_@CC composites in Figure 3 show that C, Mn, and O were the dominant elements in the samples. In the C1s spectrum, three peaks located at 284.7 eV, 285.1 eV, and 288.5 eV corresponded to the C–C/C=O bond, C–O bond, and C=O band, respectively. In addition, the XPS spectrum of O 1s (Figure 3c,e) was resolved into three components, indicating the coexistence of a Mn–O–Mn bond (529.9 eV), Mn–O–H bond (531.2 eV), and H–O–H bond (532.2 eV). Two obvious peaks appeared in the Mn 2p spectra (Figure 3f) at 653.9 eV and 642.1 eV that corresponded to Mn 2p_3/2_ and Mn 2p_1/2_, respectively. The spin-energy separation of the two peaks was 11.8 eV, further proving that the Mn oxidation state in the MnO_2_@AC and MnO_2_@CC composite was Mn^4+^ (MnO_2_) [40].

### 3.3. Electrochemical Studies

#### 3.3.1. Capacitance Performance of MnO_2_@AC

The electrochemical properties of the AC and MnO_2_@AC electrodes were studied using a three-electrode system in a 1 M Na_2_SO_4_ aqueous solution. The AC and MnO_2_@AC electrodes both possessed good reversibility. As shown in Figure 4a, the CV curves of both AC and MnO_2_@AC electrodes exhibit a rectangular shape, indicating an ideal capacitive behavior [41,42,43,44]. The integral area of CV curves of MnO_2_@AC-2 h was larger than other composites, implying the biggest electrochemical capacitance. It is noteworthy that the CV curves of MnO_2_@AC-2 h deviate to a certain extent and the electrochemical impedance increased due to the high mass loading of MnO_2_. In addition, the resistance–capacitance constant is one of the reasons why the deviation appeared. Moreover, the nearly triangular GCD (Figure 4b) of AC and MnO_2_@AC electrode indicate a high coulombic efficiency and an excellent electrochemical reversibility. As shown in Figure 4c, the GCD curves also exhibit quasi-triangular shapes, even at a high current density of 20 mA cm^−2^, which means the MnO_2_@AC electrode had good rate capability. The current response was enhanced when the mass loading of MnO_2_ was increasing, which indicated that the areal specific capacitance of AC increased for a certain degree after MnO_2_ deposition but not significantly, probably due to the blockage of the pores. It can be seen that the specific capacitance of MnO_2_@AC-5 min increased by 5.1% at 1 mA cm^−2^ and that of MnO_2_@AC-2 h increased by only 62.7%, 29.9%, and 12.2% compared to AC at a current density of 1, 10, and 20 mA cm^−2^, respectively. It even decreased at a current density of 30 mA cm^−2^ (Figure 4d). The maximum specific capacitance of MnO_2_@AC reached 1281.3 mF cm^−2^ with a 2 h deposition.

#### 3.3.2. Capacitance Performance of MnO_2_@CC

The three-electrode electrochemical properties of the CC and MnO_2_@CC electrodes are shown in Figure 5a. The CV loops of MnO_2_@CC electrode show quasi-rectangular shapes, suggesting an ideal pseudocapacitive behavior. This can be proved again via the triangular and symmetrical linear GCD curves (Figure 5b). Furthermore, as shown in Figure 5c, quasi-triangular shapes and a small IR drop (ohmic potential loss) of the GCD curves at different current densities indicates that the MnO_2_@CC electrode had a small electric resistance and a good rate stability, respectively. The electrodeposited MnO_2_@CC-10 h had a very high area specific capacitance (Figure 5d), which could reach up to 4475, 4293, 3660, and 2425 mF cm^−2^ at a current density of 1, 5, 10, 20, and 30 mA cm^−2^, respectively. According to our knowledge, this is one of the highest values ever reported for MnO_2_-based materials (Appendix A) [45,46,47,48,49,50,51,52,53]. At the same time, compared with CC, the area-specific capacitance of MnO_2_@CC could be significantly increased by 8.97, 17.9, 22.4, 22.4, and 16.6 times, which was much higher than that of AC. Based on the superior area-specific capacitance, we chose the MnO_2_@CC-10 h as the cathode to construct the ASCs.

#### 3.3.3. Capacitance Performance of ASC

Sandwich-like ASCs were fabricated using AC as an anode, MnO_2_@CC as a cathode, membrane filter as a separator, and PVA/LiCl gel as an electrolyte. AC was selected to balance the charge of MnO_2_@CC electrode for its high capacitance value in the negative potential window. Figure 6a shows that the cell voltage of the ASCs could be extended up to 1.8 V by combining the opposite potential ranges of the AC anode and MnO_2_@CC cathode. The assembled ASCs show an ideal capacitive behavior, whose CV loops shapes still remained rectangular, even at the high scan rates (Figure 6b). At different current loads, the CV curves of ASCs still show quasi-rectangular shapes and the galvanostatic charge–discharge curves (Figure 6c) are symmetrical and linear, which suggests the charges in ACSs could be transported quickly and the rate capability was good. Figure 6d shows the negative correlation between the scan rate and the area-specific capacitance of ACSs. A high areal capacitance of 3455.6 mF cm^−2^ could be obtained at a current density of 1 mA cm^−2^. 

Moreover, the ASCs assembled by AC and MnO_2_@CC had the advantages of good environmental friendliness, good biocompatibility, and low production cost. In addition, the unique multi-channel design of ASCs endowed it with very high energy and power density: the maximum energy density could be 1.56 mW h cm^−2^ when the power density was 900 mW cm^−2^ and the maximum power density was 18,000 mW cm^−2^ when the energy density was 0.249 mW h cm^−2^ (Figure 6e). Furthermore, the cycling stability of the ASCs was investigated using the galvanostatic charge–discharge texting over a potential of −1 and 0.8 V in the land-battery texting system (Figure 6f). After 10,000 cycles, the specific capacitance retention ratio of the as-fabricated asymmetric supercapacitor was still over 99%, displaying an excellent long-term cycling durability. Both electrochemical double-layer capacitance and pseudocapacitance behaviors showed up in the ASC. Therefore, the MnO_2_@CC material demonstrated a great potential as a power source in high-performance energy storage devices. 

## 4. Conclusions

In summary, we have prepared two corncob-derived carbon electrode materials with different pore structures and brought some new understanding regarding the MnO_2_-functionalized biomass carbon materials. We found that the effect of MnO_2_ highly depends on the architecture of the carbon materials. For those composed by micropores with big surface area, the deposition of MnO_2_ was fast, and easily led to clogged microstructures. On the contrary, the less-processed biomass carbon materials with a natural architecture of raw materials made a better precursor for MnO_2_ deposition. Coupling with the synergistic effects of CC and MnO_2_, the obtained MnO_2_@CC had ultrahigh area-specific capacitance and the ASCs exhibited high energy density and good cycle stability. This work provides a new strategy to effectively improve the electrochemical performance of biomass electrode materials via the electrodepositing of MnO_2_.

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
