# Peer review of "MnO2@Corncob Carbon Composite Electrode and All-Solid-State Supercapacitor with Improved Electrochemical Performance"

_materials, 2019, doi:10.3390/ma12152379_

Reviewer 1 Report

This manuscript investigates and interesting topic; however, this manuscript can be improved:

1.       Do not use contractions in academic writing (for example, “didn’t” instead “did not”); please revise the all manuscript and remove all contractions

2.       Please remove all pronouns for your manuscript.

3.       Rewrite sentence lines 51-53 – “It was interesting…”

4.       Line 72 – what is the concentration of KOH solution? What is the meaning of “(w(KOH)/weight(carbon)) = 0.5”?

5.       Why did you select that pyrolysis and carbonization conditions?

6.       Why different corncob sizes for carbonization and pyrolysis?

7.       What are the surface areas of the two materials before MnO2 deposition? What is the surface area of the pyrolyzed corncon without activation?

Reviewer 2 Report

This paper describes two corncob-derived carbon electrode materials.  The authors examined electrochemical performance after anodic electrodeposition of manganese dioxide and found the effect of manganese dioxide highly depends on the architecture of the carbon materials. I think that they carefully characterized the resulting composite electrodes and electrochemical characterization is no problem.  This work will give useful information in the field of biomass-derived functional materials.  I would like to accept this manuscript in Materials.

May I have some comments.
- Some readers may be unfamiliar with activation by KOH.  I am also curious about the activation mechanism by KOH treatment.
- The information regarding PVA (supplier, molecular weight, degrees of saponification) is missing.
